# Dilated Composite Odontoma in a Mesiodens

**DOI:** 10.3390/diagnostics15182335

**Published:** 2025-09-15

**Authors:** Aakriti Chandra, Nilima Thosar, Ramakrishna Yeluri, Ishani Rahate, Dhruvi Solanki

**Affiliations:** Department of Pediatric and Preventive Dentistry, Datta Meghe Institute of Higher Education and Research, Wardha 442001, Maharashtra, India; nilima.pedo@dmiher.edu.in (N.T.); ramakrishna.pedo@dmiher.edu.in (R.Y.); ishanirahate@gmail.com (I.R.); dhruviraj10@gmail.com (D.S.)

**Keywords:** anterior teeth, dens in dente, mesiodens, supernumerary teeth

## Abstract

Dilated Composite Odontoma also known as Dens invaginatus, “dens in dente”, or “tooth within tooth” is a rare dental anomaly resulting from enamel organ infolding during tooth development, often leading to complications like caries and pulp infection. With a prevalence of 7.45%, it commonly affects upper lateral incisors, predominantly as a Type I morphology. Mesiodens, a supernumerary tooth in the anterior maxillary midline, occurs in 89.7% of single cases and 10.3% of bilateral cases. The coexistence of dens invaginatus in a mesiodens is extremely rare, posing diagnostic and treatment challenges. This report presents a unique case of dentin invagination in a mesiodens.

**Figure 1 diagnostics-15-02335-f001:**
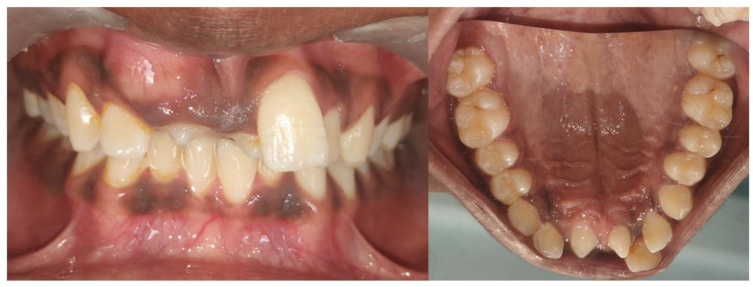
Bilateral mesiodens in the maxillary anterior region. Bilateral mesiodens is a rare entity; studies suggest that both environmental influences and genetic factors can heighten the activity of the dental lamina, resulting in the formation of an additional tooth or teeth [1]. A 14-year-old male patient accompanied by his mother was admitted to the department of pediatric dentistry with a primary complaint of unaesthetic appearance in the upper front tooth region of the jaw and wanted to have this corrected. Upon clinical examination, it was found that the right maxillary central incisor was clinically absent, indicating possible impaction. Malalignment of the anterior teeth with crowding and spacing was observed. Mild rotation of upper left central incisor teeth with a noticeable deviation of the midline was seen. Also, bilateral mesiodens were noted in the maxillary midline region, positioned palatally with partially erupted crowns. The patient did not provide any relevant medical history associated with mesiodens. Mesiodens is the most common type of supernumerary tooth, occurring in the anterior maxillary midline, and is classified as either eumorphic (resembling a normal incisor) or dysmorphic (having an irregular shape). These additional teeth often remain unerupted and can cause dental complications such as impaction, delayed eruption of permanent teeth, and crowding [2]. According to the literature, it has been reported that there is 89.7% prevalence for single mesiodens and 10.3% for bilateral cases [3]. The present case exhibits dental malalignment attributed to the presence of bilateral mesiodens that has contributed to the impaction of the right maxillary central incisor and resulted in crowding and the rotation of the adjacent teeth. Such malalignment may hinder esthetics, function, and proper oral hygiene. Management typically involves surgical extraction of the mesiodens, followed by orthodontic intervention to guide the eruption and alignment of the impacted or displaced teeth. Thus, an early diagnosis through clinical examination and radiographic evaluation is crucial to prevent long-term complications.

**Figure 2 diagnostics-15-02335-f002:**
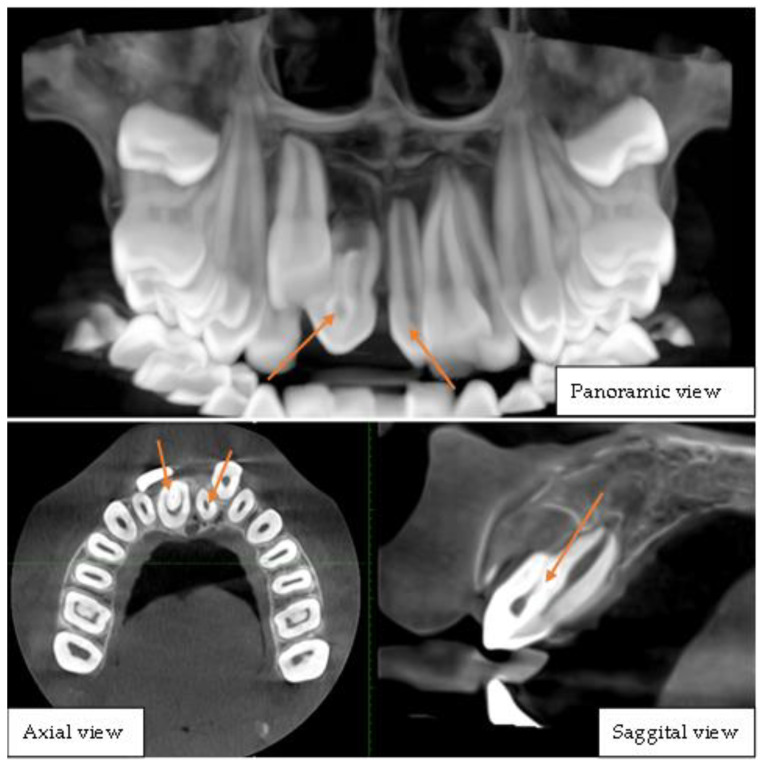
Cone-beam computed tomography of the upper jaw depicting bilateral mesiodens. This CBCT image set comprises panoramic, axial, and sagittal views, providing a comprehensive visualization of dental anomalies in the anterior maxillary region. The panoramic view reveals two mesiodens located in the midline, positioned palatally between the maxillary central incisors, one of which exhibits distinct radiographic features indicative of dens invaginatus, characterized by an infolding of enamel and dentin extending into the pulp chamber, presenting as a radiopaque structure within the tooth. The impacted right central incisor was displaced superiorly and was possibly hindered in its eruption path by the presence of these supernumerary teeth. The axial view clearly shows the mesiodens’ positions relative to the adjacent teeth, confirming their bilateral presence and the invagination pattern within one of them. The sagittal view provides a detailed cross-sectional visualization of the invaginated mesiodens, illustrating the extent of the infolding and its proximity to the pulp chamber. These findings collectively indicate a rare case of bilateral mesiodens with one exhibiting dens invaginatus, which may pose risks of dental complications such as pulp involvement, caries, or periapical pathology. Prompt intervention is crucial to manage the impacted tooth and associated anomalies. Dens invagination is a rare dental anomaly that results from an infolding of the enamel organ into the dental papilla during tooth development characterized by an invagination within the tooth structure [4,5]. Dens invaginatus has a prevalence of 7.45%, with the upper lateral incisor most affected (5.12% unilaterally) and type I as the most common morphology [6,7]. The simultaneous presence of dens invaginatus and mesiodens is exceedingly uncommon, with only a handful of cases reported in the literature [8,9,10]. The rarity of this combination makes its diagnosis and management particularly challenging. This report highlights the rarity of the rare case of dentin invagination in a mesiodens tooth which was later extracted following the treatment plan.

**Figure 3 diagnostics-15-02335-f003:**
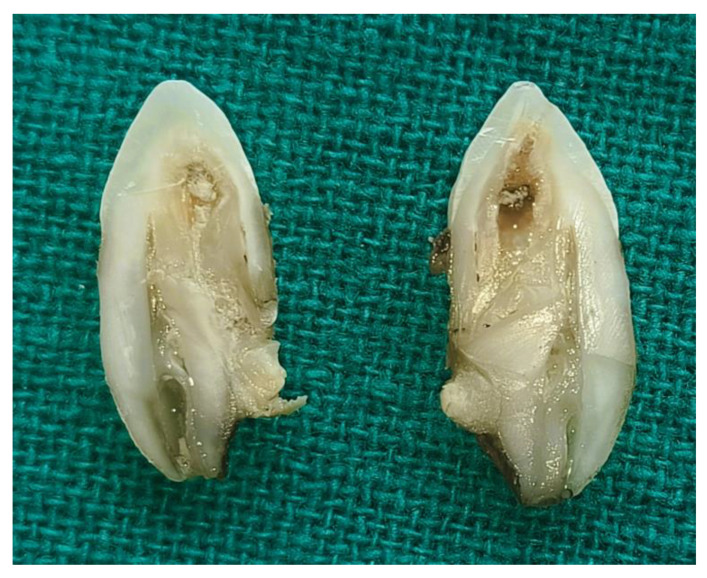
Longitudinal section of extracted mesiodens with notable dens invaginatus. Following the treatment plan, the teeth were extracted under aseptic conditions. The tooth with dens invaginatus was then sectioned longitudinally into two halves. This image displays the extracted supernumerary teeth with prominent features characteristic of dens invaginatus, showing a distinct enamel-lined infolding that extends deeply from the crown towards the root. The internal structure reveals an extensive invagination penetrating into the pulp chamber, with visible signs of debris and discoloration, suggesting a history of bacterial invasion or carious involvement. The presence of a well-defined invagination crossing the cementoenamel junction (CEJ) and extending deeper into the root structure strongly aligns with Oehlers Type II or Type III classification. In Type II, the invagination extends past the CEJ into the root but remains confined within the pulp chamber without any communication with the periodontal ligament. Conversely, Type III is characterized by an invagination that extends through the root, forming a second apical foramen that communicates directly with the periodontal ligament [4]. Based on the observed deep extension and the complexity of the invaginated structure, this case is more suggestive of Oehlers Type II, which poses a high risk of pulp infection, necrosis, and endodontic complications due to bacterial entrapment in the invaginated enamel-lined channel. This complex dental anomaly poses a considerable challenge in diagnosis and management, often necessitating advanced imaging techniques and meticulous endodontic or surgical intervention.

## Data Availability

The original data presented in this study is included in the article. Further inquiries can be directed to the corresponding author.

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
