# Peer review of "Dilated Composite Odontoma in a Mesiodens"

_diagnostics, 2025, doi:10.3390/diagnostics15182335_

Round 1

Reviewer 1 Report

Comments and Suggestions for Authors

Dear Editor, 

Kindly note, the comments are mentioned in the pdf. 

Author Response

Comment 1: Put the keywords in alphabetical order

Response 1: Thank you for pointing this out. I agree with this review hence I have made the exact changes  in the  (line no 18) keyword section of the manuscript as required - "Anterior teeth, Dens in Dente, Mesiodens, Supernumerary teeth"

comment 2 : Figure 1 , Kindly add an introduction before case description. also mention any syndrome associated with the anomalies

response 2 : we have accordingly modified the manuscript to emphasize this point in figure 1 (lines 21, 22, 23)  - "Bilateral Mesiodens is a rare entity which according to studies suggest that both environmental influences and genetic factors can heighten the activity of the dental lamina, resulting in the formation of an additional tooth or teeth [1]. " Since there are no syndrome associated with the anomalies hence we have not mentioned it.

comment 3 : kindly modiify this sentence to get a better clarity

response 3 : thank you for your comment . we have accordingly modified  the last line of figure 2 of the manuscript to get a better clarity (from line 85 - 87) "This report focuses on the uncommon instance of dentin invagination in a mesiodens tooth that was later extracted after the prescribed course of therapy."

comment 4: kindly add a note on treatment and management aspect

response 4: thank you for your comment , we agree with the correction required hence have made the necessary changes about the treatment plan in the manuscript (line no 102 & 103) - " Following the treatment plan the teeth were extracted under aseptic conditions. The tooth with dens invaginatus was then sectioned longitudinally into 2 halves. "

Reviewer 2 Report

Comments and Suggestions for Authors

Dear Authors,

Your manuscript presents a rare and clinically significant case of dens invaginatus in bilateral mesiodens—an exceptionally uncommon finding. This case is particularly relevant to pediatric dentistry, as early diagnosis of such anomalies can greatly influence treatment planning and improve long-term outcomes during a child’s growth period. The report is clearly written and well supported by clinical and radiographic findings. However, some aspects could be improved to enhance the clarity, scientific value, and clinical utility of the manuscript.

Recommendations for improvement:

  1. Figure enhancement:

To enhance the clarity and diagnostic value of the image for the reader, please include arrows in Figure 2 to highlight the specific features of the invaginated mesiodens.

  1. Redundancy in text (Lines 71-73 and 76-78):

Please consider revising these lines to eliminate redundancy, as both sections repeat similar information regarding the clinical implications of dens invaginatus. A more concise, integrated explanation would improve readability.

  1. Follow-up information:

Did the patient return for a post-extraction follow-up? If so, it would strengthen the manuscript to include clinical or radiographic images from the follow-up period, as well as details regarding any subsequent orthodontic treatment.

  1. In the References section:

Please follow the styles recommended for MDPI journals.

Author Response

comment 1: To enhance the clarity and diagnostic value of the image for the reader, please include arrows in Figure 2 to highlight the specific features of the invaginated mesiodens.

response 1: thank you for your comment, i have included the arrows in the figure 2 of the manuscript to highlight the specific feature of invaginated mesiodens

comment 2 : Redundancy in text (Lines 71-73 and 76-78):

Please consider revising these lines to eliminate redundancy, as both sections repeat similar information regarding the clinical implications of dens invaginatus. A more concise, integrated explanation would improve readability.

response 2: thank you for your comment. I have made the necessary changes in the manuscript by modifying it in more conciise sentence to reduce its duplicacy and implrove readability  (lines 73-79)- "These findings collectively indicate a rare case of bilateral mesiodens with one exhibiting dens invaginatus, which may pose risks of dental complications such as pulp involvement, caries, or periapical pathology. Prompt intervention is crucial to manage the impacted tooth and associated anomalies. Dens invagination is a rare dental anomaly that results from an infolding of the enamel organ into the dental papilla during tooth development characterised by an invagination within the tooth structure "

comment 3 : Follow-up information:

Did the patient return for a post-extraction follow-up? If so, it would strengthen the manuscript to include clinical or radiographic images from the follow-up period, as well as details regarding any subsequent orthodontic treatment.

Response 3: Thank you for your comment . the patient did not return for post extraction follow-up. Hence have not added the follow-up details.

Comment 4 : In the References section:

Please follow the styles recommended for MDPI journals. (135, 143)

response 4: thank you for your comment . i have edited the refrences according to style recommended by mdpi journal.